# Koopman Neural Forecaster for Time Series with Temporal Distribution Shifts

**Rui Wang**[*]  **Yihe Dong**  **Sercan Ö. Arik**  **Rose Yu**
UC San Diego  Google Cloud AI  Google Cloud AI  UC San Diego

## Abstract

Temporal distributional shifts, with underlying dynamics changing over time, frequently occur in real-world time series, and pose a fundamental challenge for deep neural networks (DNNs). In this paper, we propose a novel deep sequence model based on the Koopman theory for time series forecasting: Koopman Neural Forecaster (`KNF`) that leverages DNNs to learn the linear Koopman space and the coefficients of chosen measurement functions. `KNF` imposes appropriate inductive biases for improved robustness against distributional shifts, employing both a global operator to learn shared characteristics, and a local operator to capture changing dynamics, as well as a specially-designed feedback loop to continuously update the learnt operators over time for rapidly varying behaviors. We demonstrate that `KNF` achieves the superior performance compared to the alternatives, on multiple time series datasets that are shown to suffer from distribution shifts. We open-source our code at `https://github.com/google-research/google-research/tree/master/KNF`.

## 1 Introduction

Temporal distribution shifts frequently occur in real-world time-series applications, from forecasting stock prices to detecting and monitoring sensory measures, to predicting fashion trend based sales. Such distribution shifts over time may due to the data being generated in a highly-dynamic and non-stationary environment, abrupt changes that are difficult to predict, or constantly evolving trends in the underlying data distribution (Gama et al., 2014).

Temporal distribution shifts pose a fundamental challenge for time-series forecasting (Kuznetsov & Mohri, 2020). There are two scenarios of distribution shifts. When the distribution shifts only occur between the training and test domains, meta learning and transfer learning approaches (Jin et al., 2021; Oreshkin et al., 2021) have been developed. The other scenario is much more challenging: distribution shifts occurring continuously over time. This scenario is closely related to "concept drift" (Lu et al., 2018) and non-stationary processes (Dahlhaus, 1997) but has received less attention from the deep learning community. In this work, we focus on the second scenario.

To tackle temporal distribution shifts, various statistical estimation methods have been studied, including spectral density analysis (Dahlhaus, 1997), sample reweighting (Bennett & Clarkson, 2022; McCarthy & Jensen, 2016) and Bayesian state-space models (West & Harrison, 2006). However, these methods are limited to low capacity auto-regressive models and are typically designed for short-horizon forecasting. For large-scale complex time series data, deep learning models (Oreshkin et al., 2021; Woo et al., 2022; Tonekaboni et al., 2022; Zhou et al., 2022) now increasingly outperform traditional statistical methods. Yet, most deep learning approaches are designed for stationary time-series data (with i.i.d. assumption), such as electricity usage, sales and air quality, that have clear seasonal and trend patterns. For distribution shifts, DNNs have been shown to be problematic in forecasting on data with varying distributions (Kouw & Loog, 2018; Wang et al., 2021).

DNNs are black-box models and often require a large number of samples to learn. For time series with continuous distribution shifts, the number of samples from a given distribution is small, thus DNNs would struggle to adapt to the changing distribution. Furthermore, the non-linear dependencies in a DNN are difficult to interpret or manipulate. Directly modifying the parameters based on the change in dynamics may lead to undesirable effects (Vlachas et al., 2020). Therefore, if we can

---

[*]Work done during the internship at Google Cloud AI.

reduce non-linearity and simplify dynamics modeling, then we would be able to model time series in a much more interpretable and robust manner. Koopman theory (Koopman, 1931) provides convenient tools to simplify the dynamics modeling. It states that any nonlinear dynamics can be modeled by a *linear* Koopman operator acting on the space of measurement functions (Brunton et al., 2021), thus the dynamics can be manipulated by simply modifying the Koopman matrix.

In this paper, we propose a novel approach for accurate forecasting for time series with distribution shifts based on Koopman theory: Koopman Neural Forecaster (KNF). Our model has three main features: 1) we combine predefined measurement functions with learnable coefficients to introduce appropriate inductive biases into the model. 2) our model employs both global and local Koopman operators to approximate the forward dynamics: the global operator learns the shared characteristics; the local operator captures the local changing dynamics. 3) we also integrate a feedback loop to cope with distribution shifts and maintain the model's long-term forecasting accuracy. The feedback loop continuously updates the learnt operators over time based on the current prediction error.

Leveraging Koopman theory brings multiple benefits to time series forecasting with distribution shifts: 1) using predefined measurement functions (e.g., exponential, polynomial) provide sufficient expressivity for the time series without requiring a large number of samples. 2) since the Koopman operator is linear, it is much easier to analyze and manipulate. For instance, we can perform spectral analysis and examine its eigenfunctions, reaching a better understanding of the frequency of oscillation. 3) Our feedback loop makes the Koopman operator adaptive to non-stationary environment. This is fundamentally different from previous works that learns a single and fixed Koopman operator (Han et al., 2020; Takeishi et al., 2017; Azencot et al., 2020).

In summary, our major contributions include:

- Proposing a novel deep forecasting model based on Koopman theory for time-series data with temporal distributional shifts.
- The proposed approach allows the Koopman matrix to both capture the global behaviors and evolve over time to adapt to local changing distributions.
- Demonstrating state-of-the-art performance on highly non-stationary time series datasets, including M4, cryptocurrency return forecasting and sports player trajectory prediction.
- Generating interpretable insights for the model behavior via eigenvalues and eigenfunctions of the Koopman operators.

## 2 RELATED WORK

**DNNs for time-series forecasting.**   DNNs are shown to increasingly outperform traditional statistical methods (such as exponential smoothing (ETS) (Gardner Jr, 1985) and ARIMA (Ariyo et al., 2014)) for time series forecasting. For example, Tonekaboni et al. (2022); Wang et al. (2019) proposed to use DNNs to learn the local and global representations of time series separately, showing high accuracy on sales and weather data. Woo et al. (2022) leverages inductive biases in different architectures and also specially-designed contrastive loss to learn disentangled seasonal and trend representations. Sen et al. (2019) utilized a global TCN to avoid normalization before training when there are wide variations in scale. But it focuses mainly on better modeling the relationships between time series rather than advances in modeling over time as ours. Transformer-based approaches are particularly effective in time series forecasting, particularly on datasets including electricity and traffic (Zhou et al., 2022; Wu et al., 2021; Zhou et al., 2021), which are relatively stationary and have clear seasonality and trend dynamics.

**Robustness against temporal distribution shifts.**   Non-stationarity poses a great challenge for time series forecasting. To cope with varying distributions, one approach is to stationarize the input data. Kim et al. (2022) proposes a reversible instance normalization technique applied on data to alleviate the temporal distribution shift problem. Similarly, Passalis et al. (2019) utilizes a DNN to adaptively stationarize input time series. But these approaches did not improve the generalizability of DNNs. Liu et al. (2022) proposes a normalization-denormalization technique to stationarize time series, but only for transformer-based models. (Arik et al., 2022) proposes test-time adaptation with a self-supervised objective to better adapt against distribution shifts. Another line of work is to combine DNNs and statistical approaches, for better accuracy on non-stationary time series data (Makridakis et al., 2020; Malinin et al., 2021). Smyl (2020) combines ETS with a RNN, where the seasonality and

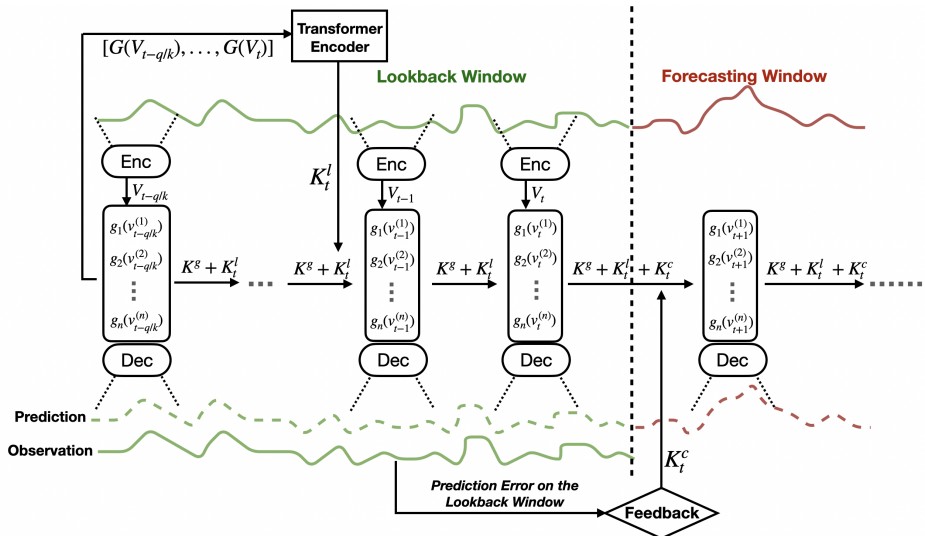

Figure 1: Architecture of the proposed KNF model. It encodes every multiple steps of observations into a measurement vector $\mathcal{G}(\boldsymbol{V}_t) = [g_1(v_1^{(1)}), ..., g_n(v_t^{(n)})]$. $\mathcal{G}(\boldsymbol{V}_t)$ is computed based on the set of predfined measurement functions $\mathcal{G}$ and their input values $\boldsymbol{V}_t$ learnt by the encoder. The model utilizes a global operator $\mathcal{K}^g$ and an adaptive local operator $\mathcal{K}_t^l$ learnt by a Transformer encoder to model the evolution of measurements. On the forecasting window, an additional adjustment operator $\mathcal{K}_t^c$ is learnt by a MLP module based on the prediction error on the lookback window.

smoothing coefficients, are fitted concurrently with the RNN weights using the same gradient descent optimization. Oreshkin et al. (2019) proposes a deep residual architecture for univariant time series data, in which each block used MLP to learn coefficients of basis functions, such as polynomials and sine functions. Indeed, we show that our model outperforms both Smyl (2020) and Oreshkin et al. (2019) on the M4 financial time series dataset (Makridakis et al., 2020).

**Koopman theory.** Koopman theory (Koopman, 1931; Strogatz, 2018) shows that a nonlinear dynamical system can be represented as an infinite-dimensional linear Koopman operator acting on a space of measurement functions. The spectral decomposition of the linear Koopman operator can provide insights on the behaviors of nonlinear systems. Traditionally, dynamic mode decomposition (DMD) (Brunton et al., 2016) is commonly used for approximating the Koopman Operator. However, it is highly nontrivial to find appropriate measurement functions as well as the Koopman operator. Many works at the intersection of machine learning and Koopman theory employ DNNs to learn measurement functions from data (Han et al., 2020; Li et al., 2020; Morton et al., 2019; Li & Jiang, 2021; Takeishi et al., 2017; Fan et al., 2022). Yeung et al. (2019) and Lusch et al. (2018) propose to use fully connected DNNs to learn the transformation between the observations and measurement functions that span a Koopman invariant subspace and the Koopman operator is approximated by a linear layer. Azencot et al. (2020) also designed an autoencoder architecture based on Koopman theory to forecast fluid dynamics. Different from these approaches, we allow the Koopman matrix to evolve over time for each time series to adapt to the changing distribution.

## 3 METHODOLOGY

### 3.1 TIME-SERIES FORECASTING WITH KOOPMAN THEORY

Time series data $\{\boldsymbol{x}_t\}_{t=1}^T$ can be considered as observations of a dynamical system states – consider following discrete form: $\boldsymbol{x}_{t+1} = \boldsymbol{F}(\boldsymbol{x}_t)$, where $\boldsymbol{x} \in \mathcal{X} \subseteq \mathbb{R}^d$ is the system state, and $\boldsymbol{F}$ is the underlying governing equation. We focus on multi-step forecasting task of predicting the future states given a sequence of past observations. Formally, we seek a function map $f$ such that:

$$f : (\boldsymbol{x}_{t-q+1}, \dots, \boldsymbol{x}_t) \longrightarrow (\boldsymbol{x}_{t+1}, \dots, \boldsymbol{x}_{t+h}), \tag{1}$$

where $q$ is the lookback window length and $h$ is the forecasting window length.

Koopman theory (Koopman, 1931) shows that any nonlinear dynamic system can be modeled by an infinite-dimensional linear Koopman operator acting on the space of all possible measurement functions. More specifically, there exists a linear infinite-dimensional operator $\mathcal{K} : \mathcal{G}(\mathcal{X}) \mapsto \mathcal{G}(\mathcal{X})$ that acts on a space of real-valued measurement functions $\mathcal{G}(\mathcal{X}) := \{g : \mathcal{X} \mapsto \mathbb{R}\}$. The Koopman operator maps between function spaces and advances the observations of the state to the next step:

$$\mathcal{K}g(\boldsymbol{x}_t) = g(\boldsymbol{F}(\boldsymbol{x}_t)) = g(\boldsymbol{x}_{t+1}). \qquad (2)$$

We propose Koopman Neural Forecasting (KNF), a deep sequence model based on Koopman theory to forecast highly non-stationary time series, as shown in Fig. 1. It instantiates an encoder-decoder architecture for each time series segment. The encoder takes in observations from multiple time steps as the underlying dynamics may contain higher-order time derivatives. Our model has three main features: 1) we use predefined measurement functions with learned coefficients to map time series to the functional space. 2) the model employs both a global Koopman operator to learn the shared characteristics and a local Koopman operator to capture the local changing dynamics. 3) we also integrate a feedback loop to update the learnt operators over time based on forecasting error, maintaining model's long-term forecasting performance.

## 3.2 Leveraging Predefined Measurement Functions

We define a set of measurement functions $\mathcal{G} := [g_1, \cdots, g_n]$ that spans the Koopman space, where each $g_i : \mathbb{R} \mapsto \mathbb{R}$. For example, $g_1(x) = \sin(x)$. These functions are canonical nonlinear functions and are often used to model complex dynamical systems, such as Duffing oscillator and fluid dynamics (Brunton et al., 2021; Kutz et al., 2016). They also provide a sample-efficient approach to represent highly nonlinear behavior that may be difficult to learn for DNNs.

We use an encoder to generate the coefficients of the measurement functions $\Psi(\boldsymbol{X}_t)$, such as the frequencies of sine functions. Let $n$ be the number of measurement functions for each feature, $d$ be the number of features in a time series and $k$ be the number of steps encoded by the encoder $\Psi : \mathbb{R}^{d \times k} \mapsto \mathbb{R}^{n \times d \times k}$ every time. The lookback window length $q$ is a multiple of $k$ and we denote $\boldsymbol{x}_{tk:(t+1)k}$ as $\boldsymbol{X}_t \in \mathbb{R}^{d \times k}$ for simplicity.

As shown in the Eq.3 below, we first obtain a latent matrix $\boldsymbol{V}_t = [\boldsymbol{v}_t^{(1)}, \boldsymbol{v}_t^{(2)}, \cdots, \boldsymbol{v}_t^{(n)}] \in \mathbb{R}^{n \times d}$. Every vector $\boldsymbol{v_i} \in \mathbb{R}^d$ is a different linear transformation of the observations, where the weights are learnt by the encoder $\Psi$:

$$\boldsymbol{V}_t[i,j] = \sum_l \Psi(\boldsymbol{X}_t)[i,j,l]\boldsymbol{X}_t[j,l]; \quad 1 \le i \le n,\ 1 \le j \le d,\ 1 \le l \le k. \qquad (3)$$

Our measurement functions are defined in the latent space rather than the observational space. We apply a set of predefined measurement functions $\mathcal{G}$ to the latent matrix $\boldsymbol{V}_t$:

$$\mathcal{G}(\boldsymbol{V}_t) = [g_1(\boldsymbol{v}_t^{(1)}), g_2(\boldsymbol{v}_t^{(2)}), ..., g_n(\boldsymbol{v}_t^{(n)})] \in \mathbb{R}^{n \times d} \qquad (4)$$

In our implementation, we flatten $\mathcal{G}(\boldsymbol{V}_t)$ into a vector and then finite Koopman operator should be a $nd \times nd$ matrix. Finally, we use a decoder $\Phi : \mathbb{R}^{n \times d} \mapsto \mathbb{R}^{k \times d}$ to reconstruct the observations from the measurements:

$$\hat{\boldsymbol{X}}_t = \Phi(\mathcal{G}(\boldsymbol{V}_t)). \qquad (5)$$

Here, the encoder $\Psi$ and the decoder $\Phi$ can be any DNN architecture, for which we use multi-layer perceptron (MLP). The set of measurement functions $\mathcal{G}$ contains polynomials, exponential functions, trigonometric functions as well as interaction functions. These predefined measurement functions are useful in imposing inductive biases into the model and help capture the non-linear behaviors of time series. The encoder model needs to approximate only the parameters of these functions without the need of directly learning non-stationary characteristics. Ablation studies (in Sec. 4.8) demonstrate that using predefined measurement functions significantly outperforms the model with learned measurement functions in the previous works.

## 3.3 Global and Local Koopman Operators

Dynamic mode decomposition (DMD) (Tu et al., 2013) is traditionally used to find the Koopman operator that best propagates the measurements. But for time series with temporal distribution shift,

we need to compute spectral decomposition and learn a Koopman matrix for every sample (i.e. slice of a trajectory), which is computationally expensive. So we utilize DNNs to learn Koopman operators.

In classic Koopman theory, the measurement vector $\mathcal{G}(\boldsymbol{V}_t)$ is infinite-dimensional, which is impossible to learn. We assume that encoder is learning a finite approximation and $\mathcal{G}(\boldsymbol{V}_t)$ forms a finite Koopman-invariant subspace. Thus, the Koopman operator $\mathcal{K}$ that we need to find is the finite matrix that best advances the measurements forward in time.

While the Koopman matrix should vary across samples and time in our case, it should also capture the global behaviors. Thus, we propose to use both a global operator $\mathcal{K}^g$ and a local operator $\mathcal{K}_t^l$ to model the propagation of dynamics in the Koopman space, defined as below:

$$\mathcal{K}\mathcal{G}(\boldsymbol{V}_t) := (\mathcal{K}^g + \mathcal{K}_t^l)\mathcal{G}(\boldsymbol{V}_t) = \mathcal{G}(\boldsymbol{V}_{t+1}), \quad t \geq 0. \tag{6}$$

The global operator $\mathcal{K}^g$ is an $nd \times nd$ trainable matrix that is shared across all time series. We use the global operator to learn the shared behavior such as trend. The local Koopman operator $\mathcal{K}_t^l$, on the other hand, is based on the measurement functions on the lookback window for each sample, shown in Fig. 1. The local operator should capture the local dynamics specific to each sample. Since we generate the forecasts in an autoregressive way, the local operator depends on time $t$ and varies across autoregressive steps, adapting to the distribution changes along prediction. We use a Transformer architecture with a single-head as the encoder, to capture the relationships between measurements at different steps. We use the attention weight matrix in the last layer as the local Koopman operator.

## 3.4 FEEDBACK LOOP

Suppose an abrupt distributional shift occurs in the middle of the look-back window, the model would still try to fit two distributions before and after the shift but a single proporgration matrix is never good enough to model multiple distributions. This will results in the inaccurate operator used for the forecasting window. To address it, we add an additional feedback closed-loop, in which we employ an MLP module $\Gamma$ to learn the adjustment operator $\mathcal{K}_t^c$ based on the prediction errors in the lookback window. It is directly added to other operators when making predictions on the forecasting window, as shown in Fig. 1. More specifically, we apply global and local operators recursively to the measurements at the first step in the lookback window to obtain predictions:

$$\hat{\boldsymbol{X}}_{t-q/k+i} = \Phi((\mathcal{K}^g + \mathcal{K}_t^l)^i \mathcal{G}(\boldsymbol{V}_{t-q/k})), \quad 0 < i \leq q/k. \tag{7}$$

Then, the feedback module $\Gamma$ uses the difference between the predictions on the lookback window and the observed data to generate additional adjustment operator $\mathcal{K}_t^c$, which is a diagonal matrix:

$$\mathcal{K}_t^c = \Gamma(\hat{\boldsymbol{X}}_{t-q/k:t} - \boldsymbol{X}_{t-q/k:t}) = \Gamma(\hat{\boldsymbol{x}}_{t-q:t} - \boldsymbol{x}_{t-q:t}) \tag{8}$$

If the predictions deviate significantly from the ground truth within the lookback window, the operator $\mathcal{K}_t^c$ would learn the temporal change in the underlying dynamics and correspondingly adjust the other two operators. Thus, for forecasting, the sum of all three operators is used:

$$\hat{\boldsymbol{X}}_{t+i} = \Phi((\mathcal{K}^g + \mathcal{K}_t^l + \mathcal{K}_t^c)^i \mathcal{G}(\boldsymbol{V}_t)), \quad i > 0. \tag{9}$$

In a word, the feedback module is designed to detect the distributional shifts in the lookback window and adapt the global+local operator to the latest distribution in the lookback window.

## 3.5 LOSS FUNCTIONS

KNF is trained in an end-to-end fashion, using superposition of three loss terms $L = L_{\text{rec}} + L_{\text{back}} + L_{\text{forw}}$. Denote $l$ as a distance metric for which we use the L2 loss. The first term is the reconstruction loss, to ensure the decoder $\Phi$ can reconstruct the time series from the measurements:

$$L_{\text{rec}} = l(\boldsymbol{X}_t, \Phi(\mathcal{G}(\Psi(\boldsymbol{X}_t)\boldsymbol{X}_t))), \quad t \geq 0. \tag{10}$$

The second term is the prediction loss on the lookback window to ensure the sum of global and local operators is the best-fit propagation matrix on the lookback window.

$$L_{\text{back}} = l(\boldsymbol{X}_{t-q/k+i}, \Phi((\mathcal{K}^g + \mathcal{K}_t^l)^i \mathcal{G}(\Psi(\boldsymbol{X}_{t-q/k})\boldsymbol{X}_{t-q/k})), \quad 0 < i \leq q/k. \tag{11}$$

The third term is for prediction accuracy in the forecasting window to guide the feedback loop to learn the correct adjustment placed on the Koopman operator.

$$L_{\text{forw}} = l(\boldsymbol{X}_{t+i}, \Phi((\mathcal{K}^g + \mathcal{K}_t^l + \mathcal{K}_t^c)^i \mathcal{G}(\boldsymbol{V}_t)), \quad i > 0. \tag{12}$$

## 4 EXPERIMENTS

### 4.1 DATASETS

We benchmark our method on three time series datasets: M4 (Makridakis et al., 2020), Crypto (Arik et al., 2022) and Basketball Player Trajectories (Li et al., 2021). These time series are particularly chosen as they are difficult to forecast due to high nonstationarity and abrupt temporal changes, as analyzed in Sec. 4.2.

**M4.** It contains 10000 highly nonstationary univariate time series with different frequencies from hourly to yearly and different categories from financials to demographics. The forecasting horizon varies across different frequencies. We directly compare KNF with the M4 competition winner (Smyl, 2020), the second place (Montero-Manso et al., 2020) and the ensemble N-Beats-I+G (Oreshkin et al., 2019) that has achieved the competitive prediction performance on M4.

**Crypto.**[1] This multivariate dataset contains 8 features on historical trades, such as open and close prices, for 14 cryptocurrencies. The original challenge is to predict 3-step ahead 15-minute relative future returns. Since we focus on long-term forecasting, we train all models to make 15-step predictions of 15-minute relative future returns. We use the original training set from the competition and do an 80%-10%-10% training-validation-test split.

**Player Trajectory.**[2] This dataset contains basketball player movement trajectories from NBA games in 2016. We randomly sample 300 player trajectories for training and validation and 50 trajectories for testing. All models are trained to yield 30-step ahead predictions.

### 4.2 DISTRIBUTION SHIFTS AND FORECASTABILITY

As a way of motivating for the improvement scenarios of our method, we show that the three datasets we focus are much more difficult to predict than a commonly-used datasets like Electricity (Harries & Wales, 1999) with the following three metrics: (1) Forecastability (Goerg, 2013): it is one minus the entropy of Fourier decomposition of the time series; (2) Lyapunov exponents (LEs) (Dingwell, 2006; Schölzel, 2019): a measure of how sensitive a dynamical system is to initial conditions.(3) Trend: the slope of the linear regression fitted on the time series scaled by its own magnitude; and (4) Seasonality: we use ACF test (Witt et al., 1998) to test if there is clear seasonality. We report the mean forecastability, mean trend and the percentage of the slices that have seasonality in Table 1, averaged over slices with length 20. We can see that seasonality is dominant for Electricity dataset, which also has significantly higher forecastability compared to the datasets we experiment with. Moreover, we show that KNF achieves the state-of-the-art prediction performance on those datasets that have high Lyapunov exponents, low forecastability, no clear trends and seasonality, including Crypto, Player Trajectories, M4-monthly, M4-weekly and M4-daily.

|  | Electricity | **Crypto** | **Player Traj.** | **M4-monthly** | **M4-weekly** | **M4-daily** | M4-hourly | M4-yearly | M4-quarterly |
|---|---|---|---|---|---|---|---|---|---|
| Forecastability | 0.77 | 0.35 | 0.49 | 0.44 | 0.43 | 0.44 | 0.46 | 0.58 | 0.47 |
| LEs | 0.005 | 0.026 | 0.052 | 0.011 | 0.013 | 0.020 | 0.003 | 0.004 | 0.003 |
| Trend | 0.00 | 0.02 | 0.01 | 0.48 | 0.13 | 0.05 | 0.02 | 4.32 | 1.06 |
| Seasonality | 100% | 0.00% | 0.00% | 66.34% | 0.00% | 0.00% | 99.76% | 0.00% | 84.51% |

Table 1: The forecastability, Lyapunov exponents, trend and seasonality of the datasets that we used for our experiments, compared with a commonly-used Electricity benchmark. The boldface datasets are what we use for experiments in this paper.

### 4.3 BASELINES

For M4, we compare with the winner solution (Smyl, 2020), the second best (Montero-Manso et al., 2020) and N-beats (Oreshkin et al., 2019). On the Crypto and Player Trajectory datasets, we compare KNF with four different types of models.

---

[1] https://www.kaggle.com/competitions/g-research-crypto-forecasting/data
[2] https://github.com/linouk23/NBA-Player-Movements

- **Vector ARIMA** (`VARIMA`) (Stock & Watson, 2001): It is an extension of the classic ARIMA model for multivariate time series.

- **Multi-layer Perceptron** (`MLP`) that maps the historic observations to the future and roll out autoregressively to yield long-term predictions. It is a commonly-used DL model for time series forecasting (Arik et al., 2022; Wang et al., 2021; Faloutsos et al., 2018).

- **Random Forest** (`RF`) applied autoregressively same as `MLP`. It is a traditional ML model that have achieved competitive performance on many time series benchmarks (Godahewa et al., 2021; Masini et al., 2021; Kane et al., 2014; Hyndman & Athanasopoulos, 2018).

- **Long Expressive Memory** (`LEM`) (Rusch et al., 2022) : An SOTA recurrent model for learning long-term sequential dependencies.

- **FedFormer** (`FedFormer`) (Zhou et al., 2022) is a state-of-art transformer-based model, which has outperformed other transformer-based models, such as Informer (Zhou et al., 2021) and Autoformer (Wu et al., 2021), on many datasets, including electricity, traffic, weather, etc.

- **Variational Beam Search** (`VBS`) (Li et al., 2021) is a Bayesian online learning model proposed to detect and adapt to temporal distributional shifts. Since VBS is not designed for time series forecasting, we modify it by feeding its own prediction of next step back to the input (instead of the ground truth) to yield multi-step predictions.

## 4.4 TRAINING STRATEGIES

We find that the following two training strategies can improve the prediction accuracy of models by varying degrees. The first one is the reversible instance normalization `ReVIN` (Kim et al., 2022), which normalize the input sequence and de-normalize the predictions at every autoregressive step for every instance. We use ReVIN for `MLP` and `KNF` since it can improve their prediction accuracy. An ablation study of ReVIN on `KNF` can be found in Table 4. The second one is, temporal bundling `TB` (Brandstetter et al., 2022) that asks autoregressive models to generate multiple-step predictions instead of just one on every call, to reduce number of model calls and therefore error propagation speed. We observe this strategy can improve prediction accuracy of both `MLP` and `RF`.

## 4.5 SETUP

For all datasets, we use a sliding window approach to generate training samples. On Crypto and Player Trajectory datasets, we perform a grid search of hyperparameters, including learning rate, input length, hidden dimension, number of predictions made in each autoregressive step, etc. for all models. The hyperparameter tuning ranges can be found in the Appendix A.3 Table 6. The default

| sMAPE | Monthly(18) | Weekly(13) | Daily(14) |
|---|---|---|---|
| Montero et.al | 12.639 | 7.625 | 3.097 |
| Smyl | 12.126 | 7.817 | 3.170 |
| Nbeats-I+G | 12.024 | - | - |
| KNF | **11.930** | **7.254** | **2.990** |

Table 2: sMAPE for `KNF` and baselines on M4 datasets. The numbers in parentheses are the number of prediction steps. `KNF` achieves the state-of-the-art prediction performance at Weekly, Daily and Monthly frequencies.

set of measurement functions used in `KNF` includes polynomials up to the order of four, one exponential function and trigonometric functions with the same number of input steps for each feature, as well as pairwise product interaction functions between features if the time series data is multivariate.

We report the mean and standard deviation averaged across five runs. We follow the literature and use RMSE for evaluation on Player Trajectories and the weighted RMSE on Crypto, where the weight corresponds to the importance of each cryptocurrency same as in the competition. Since `VBS` is deterministic once its inverse temperature parameter is fixed, we do not report its standard deviation. On M4, we evaluate models with the sMAPE metric (Makridakis et al., 2020)[1] used in the original competition. Ensembling is used by most models in the M4-competition and N-Beats (Oreshkin et al., 2019), so we ensemble five `KNF` with best hyperparameters but trained with random seeds. Since the evaluation in M4 competition is only based on a single submission, we also report the sMAPE of a single ensembled prediction without standard deviation.

---

[1]sMAPE is the mean absolute error scaled by the magnitude of the predictions and target.

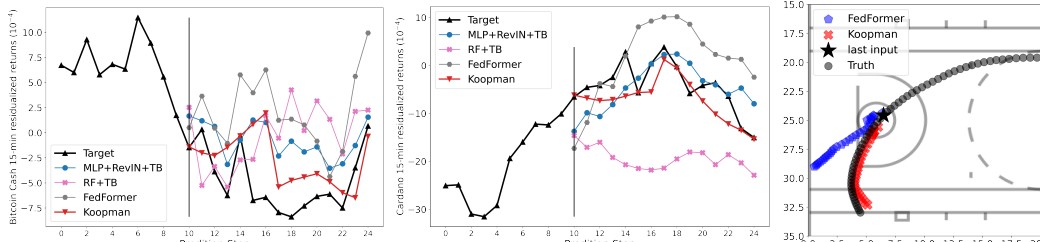

Figure 2: Left: Example forecasts on Crypto dataset. Right: Example forecasts on NBA player trajectory dataset. We observe that KNF can be superior in capturing complex non-stationary patterns.

## 4.6 RESULTS ON M4

Table 2 reports the prediction sMAPE of KNF and three baselines, including the M4 winner (Smyl, 2020), the second place (Montero-Manso et al., 2020) and the ensembled N-beats-I+G (Oreshkin et al., 2019), on six datasets with different frequency in M4. Oreshkin et al. (2019) does not report the breakdown sMAPE of N-beats on weekly, daily and hourly, so we did not include them in the table. The number behind each frequency in the table is the number of prediction steps required in the competition. We can see that KNF achieve the state-of-the-art accuracy on Weekly, Daily and Monthly data that have longer forecasting horizons, and no clear trends and seasonality. This highlights the value proposition of KNF for accurate long-term forecasting accuracy especially for time-series with nonstationary characteristics and low forecastability. KNF also achieve competitive performance on Yearly, Quarterly and Hourly and the full results on M4 are in Appendix A.4.

## 4.7 RESULTS ON CRYPTO AND PLAYER TRAJECTORIES

Table 3 shows the Prediction RMSE on Crypto and Player Trajectories datasets. KNF consistently achieves the best performance on both, across different forecasting horizons. Fig. 2 exemplifies the predictions of KNFand best-performing baselines. We observe that KNF can capture both overall trends and small fluctuations in a superior way, yielding much higher accuracy. Fig. 2 right visualizes the predictions on a trajectory with changing direction. KNF correctly predicts the change of moving direction while FedFormer fails. Regarding baseline performances, RevIN and TB greatly improves MLP, and MLP+RevIN+TB is the best performing one on Crypto, and FedFormer performs better on Player Trajectory, that are less chaotic and irregular. Though VBS was shown to perform well on the online change point detection task, it does not appear to be as successful for time series forecasting based on its performance on these two datasets. We also include additional experimental results on the Traffic and Exchange Datasets (Zeng et al., 2023) in Appendix A.5.

## 4.8 ABLATION STUDY ON M4

We performed an ablation study of KNF on the M4-Weekly data to understand the contribution of each component in our model. Table 4 shows the sMAPE of ensembled predictions from five funs of each variant. We denote KNF-base as the basic backbone of our model that only has an encoder-decoder architecture and a local Koopman operator. KNF-base-G uses purely data-driven measurements,

| Model | Crypto (Weighted RMSE $10^{-3}$) | | | | Basketball Player Trajectory (RMSE) | | | |
|---|---|---|---|---|---|---|---|---|
| | (1~5) | (6~10) | (11~15) | Total | (1~10) | (11~20) | (21~30) | Total |
| VARIMA | 6.09±0.00 | 8.83±0.00 | 10.74±0.00 | 8.76±0.00 | **0.22±0.00** | 0.90±0.00 | 1.98±0.00 | 1.26±0.00 |
| MLP | 6.68±1.53 | 7.95±0.33 | 8.64±0.55 | 7.85±0.35 | 0.73±0.20 | 1.64±0.31 | 2.77±0.42 | 1.91±0.32 |
| MLP+RevIN+TB | **5.03±0.08** | 7.16±0.13 | 8.41±0.06 | 7.01±0.08 | 0.37±0.02 | 1.16±0.03 | 2.25±0.04 | 1.48±0.25 |
| RF+TB | 6.62±1.30 | 7.99±0.24 | 8.51±1.19 | 7.84±0.04 | 0.86±0.01 | 2.10±0.01 | 3.48±0.02 | 2.40±0.01 |
| FedFormer | 5.61±0.05 | 7.50±0.03 | 8.89±0.03 | 7.46±0.04 | 0.43±0.02 | 0.92±0.03 | 1.97±0.04 | 1.29±0.03 |
| LEM | 5.27±0.02 | 7.23±0.06 | 8.23±0.05 | 7.02±0.04 | 0.33±0.01 | 1.08±0.04 | 2.18±0.02 | 1.42±0.02 |
| VBS | 15.23±0.00 | 14.46±0.00 | 26.49±0.00 | 19.52±0.00 | 0.90±0.00 | 2.84±0.00 | 9.24±0.00 | 5.60±0.00 |
| KNF | 5.24±0.00 | **7.03±0.01** | **7.63±0.01** | **6.91±0.01** | 0.26±0.01 | **0.84±0.01** | **1.81±0.01** | **1.16±0.01** |

Table 3: Prediction RMSE on Cryptos and Player Trajectories datasets.

| M4-Weekly | sMAPE |
|---|---|
| KNF-base-G | 14.175 |
| KNF-base-I | 9.122 |
| +RevIN | 8.435 |
| +RevIN+$\mathcal{K}^g$ | 7.500 |
| +RevIN+$\mathcal{K}^g$+feedback | **7.254** |

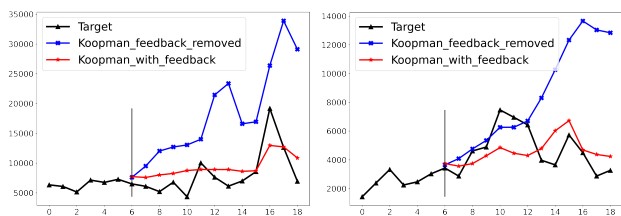

Table 4: Ablation study of KNF on M4-weekly data. It shows the importance of every component in the model architecture.

Figure 3: Predictions on M4-weekly data from original KNF and with its version without the feedback loop. We observe that the feedback loop can help providing robustness against temporal distributional shifts and maintain the long-term prediction accuracy.

which is similar to the Koopman autoencoders proposed in (Yeung et al., 2019; Azencot et al., 2020). This can be considered a baseline for our model. KNF-base-I uses predefined measurement functions, which significantly outperforms KNF-base-G. This demonstrates the benefits of leveraging hard-coded functions. Moreover, adding any of ReVIN, the global Koopman operator and the feedback loop also brings great improvement based on the results shown in Table 4 .

To further demonstrate the effectiveness of the feedback loop, we visualize the predictions on M4-weekly data from KNF and KNF with the feedback module removed after training. We can observe that the predictions from the model with the feedback loop removed start to deviate from the ground truth after a few steps. That means the feedback loop can cope with the temporal distributional shifts and thus improve the long-horizon forecasting accuracy. We perform additional ablation studies on measurement functions on M4-yearly data shown in Appendix A.1, demonstrating the necessity of each type of measurement functions.

## 4.9 INTERPRETABILITY RESULTS

KNF can offer unique interpretability capabilities via spectral analysis of the Koopman operators. We perform eigen-decomposition on $\mathcal{K}^g + \mathcal{K}^l_t$ for KNF trained on M4-weekly data to investigate what individual eigenfunctions have learnt. Fig. 4 shows the predictions in the lookback window with different eigenfunctions. From the top to the bottom, the plot shows the target, the reconstruction from KNF with only the first eigenfunction in the lookback window and the reconstruction only using the second eigenfunction. We observe that the some eigenfunctions captures the trend while some other functions focus on the seasonality.

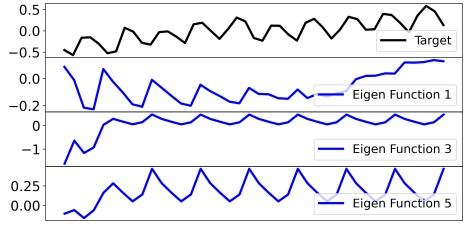

Figure 4: Reconstructions from KNF on the lookback window with only one of eigenfunctions of $\mathcal{K}^g + \mathcal{K}^l_t$ on a M4 time series.

To show that our model can also correctly learn the time series governed by known equations. We run experiments with our model on a simple oscillator system, given in Appendix A.2. We try our models with different sets of polynomial functions and train/test them on different initial conditions. Fig. 5 shows that our model can make accurate predictions on the test set with different dictionaries.

## 5 CONCLUSION

In this paper, we propose a novel model based on the Koopman theory, Koopman Neural Forecaster (KNF), designed for accurate long-term forecasting for non-stationary time series with temporal distribution shifts. KNF leverages predefined measurement functions to capture the nonlinear and nonstationary characteristics that may pose a great difficulty for neural nets to learn. It employs both a global operator to learn shared characteristics, and a local operator to capture changing dynamics. We also use a judiciously-designed feedback loop to continuously update the learnt operators over time for rapidly varying behaviors. We demonstrate that KNF achieves the state-of-the-art performance on wide range of time series datasets that are particularly known to suffer from distribution shifts. KNF achieves SOTA prediction performance on M4, Cryptos and NBA player trajectory datasets and provides interpretable results. Future work includes forecasting higher-dimensional dynamics and investigating distributional shifts between training and test domains.

## ACKNOWLEDGEMENT

We'd like to thank Tomas Pfister, Nate Yoder, and Jinsung Yoon for insightful discussions on this work. This work was supported in part by U.S. Department Of Energy, Office of Science, U. S. Army Research Office under Grant W911NF-20-1-0334, and NSF Grants #2134274, #2107256 and #2134178.

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

# A APPENDIX

## A.1 ABLATION STUDIES ON PREDEFINED MEASUREMENT FUNCTIONS

Table 5 shows the results from the ablation study of measurement function on M4-Yearly data. G represents generic, purely data-driven. P is polynomial function, E is the exponential function, S is the sine function. The integers in the first row are the number of each type of function. Basically, we gradually add more and different types of predefined measurement functions to purely data-drive model and observe improvements brought by the polynomial, exponential function, sine functions.

| KNF | G | P1 | P2 | P3 | P2+E1 | P2+E2 | P2+E3 | P2+E1+S1 | P2+E1+S2 | P2+E1+S3 | P2+E1+S4 |
|---|---|---|---|---|---|---|---|---|---|---|---|
| sMAPE | 14.66 | 14.65 | 14.56 | 14.79 | 14.48 | 14.63 | 14.66 | 14.35 | 14.16 | 14.01 | 14.15 |

Table 5: Ablation study of measurement function on M4-Yearly data. G represents generic, purely data-driven. P is polynomial function, E is the exponential function, S is the sine function. The integers in the first row are the number of each type of function.

## A.2 SIMPLE NONLINEAR SYSTEM WITH FINITE KOOPMAN SPACE

We consider the following simple nonlinear system with discrete spectrum.

$$\dot{x_1} = \mu x_1$$
$$\dot{x_2} = \lambda(x_2 - x_1^2)$$

This system has a minimal Koopman invariant subspace spanned by $\{x_1, x_2, x_1^2\}$:

$$\frac{d}{dt}\begin{bmatrix} x_1 \\ x_2 \\ x_1^2 \end{bmatrix} = \begin{bmatrix} \mu & 0 & 0 \\ 0 & \lambda & -\lambda \\ 0 & 0 & 2\mu \end{bmatrix}\begin{bmatrix} x_1 \\ x_2 \\ x_1^2 \end{bmatrix}$$

It can also be spanned by three eigenfunctions $[\phi_1, \phi_2, \phi_3] = [x_1, x_1^2, x_2 - \frac{\lambda}{\lambda-2\mu}x_1^2]$

$$\frac{d}{dt}\begin{bmatrix} \phi_1 \\ \phi_2 \\ \phi_3 \end{bmatrix} = \begin{bmatrix} \mu & 0 & 0 \\ 0 & 2\mu & 0 \\ 0 & 0 & \lambda \end{bmatrix}\begin{bmatrix} \phi_1 \\ \phi_2 \\ \phi_3 \end{bmatrix}$$

Any multiplication of $[\phi_1, \phi_2, \phi_3]$ is still an eigenfunction.

We want to show our model with only trainable global operator can correctly identify the Koopman invariant subspace from the synthetic data generated based on this system. We generate 36 time series with $\mu = -0.1$ and $\lambda = -1$ and the initial values are uniformly sampled from $[-1, 1]^2$. We tried four different basis dictionaries, including $D_1 = \{x_1, x_2, x_1^2\}$, $D_2 = \{x_1, x_2, x_1^2, x_2^2\}$, $D_3 = \{x_1, x_2, x_1^2, x_2^2, x_1^3, x_2^3\}$, $D_4 = \{x_1, x_2, x_1^2, x_2^2, x_1^3, x_2^3, x_1^4, x_2^4\}$ and we trained and tested the models on different initial conditions. The left figure in Fig. 5 shows eigenvalues learnt by KNF and some true eigenvalues. Since the learned Koopman matrix may not be unique given the flexibility in the coefficients of measurement functions, and the compositions of eigenfunctions are also eigenfunctions, there are many possible eigenvalues. But we can still see that most of the learned eigenvalues match the true eigenvalues. The right three figures in Fig. 5 shows that KNF can make accurate predictions with different dictionaries.

## A.3 ADDITIONAL EXPERIMENTAL DETAILS

Table 6 shows the hyperparameter tuning ranges, including the learning rate, the hidden dimension and number of layers of deep neural network modules in both baselines and our model, number of predictions made at each autoregressive step, the length of input observations, the forecasting window size during training, and whether to use ReVIN. For different modules in a model, we tune hyperparameters, such as the number of layers/hidden dimensions separately. For instance, in FedFormer, the encoder and decoder may have different numbers of layers. As for the nonlinearity, we use ReLU for all layers.

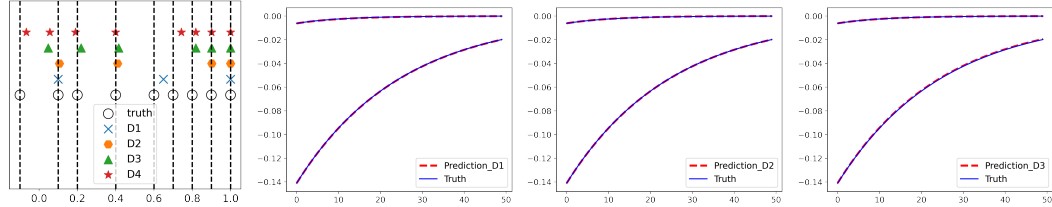

Figure 5: Left: Learnt eigenfunctions by KNF with different dictionaries. Right: Predictions from KNF with on the simple nonlinear system.

| Learning rate | Hidden dim | #Layers | #forecasting window size during training | #Predictions made at each autoregressive step | Input length | Whether to use ReVIN |
|---|---|---|---|---|---|---|
| 1e-1∼1e-5 | 64∼1024 | 3∼7 | 1∼15 | 1∼10 | 5∼50 | True/False |

Table 6: Hyperparamter Tuning Ranges.

## A.4  ADDITIONAL RESULTS

| sMAPE | Monthly(18) | Weekly(13) | Daily(14) | Hourly(48) | Yearly(6) | Quarterly(8) |
|---|---|---|---|---|---|---|
| Montero et.al | 12.639 | 7.625 | 3.097 | 11.506 | 13.528 | 9.733 |
| Smyl | 12.126 | 7.817 | 3.170 | **9.328** | 13.176 | 9.679 |
| Nbeats-I+G | 12.024 | - | - | - | **12.924** | **9.212** |
| KNF | **11.930** | **7.254** | **2.990** | 11.294 | 13.800 | 10.008 |

Table 7: sMAPE for KNF and baselines on six M4 datasets for different sampling frequencies. The numbers in parentheses are the number of prediction steps. KNF achieves the state-of-the-art prediction performance at Weekly, Daily and Monthly frequencies.

## A.5  EXPERIMENTS ON THE TRAFFIC AND EXCHANGE BENCHMARK DATASETS

We also experiment with KNF on the the Traffic and Exchange Benchmark Datasets. Our results in Table 8 demonstrate that KNF achieves MAE scores comparable to the state-of-the-art model (Zeng et al., 2023). Additionally, KNF outperforms the SOTA model in terms of MSE on the traffic dataset, indicating that KNF generates relatively stable predictions across the forecasting horizon.

| Models Metrics | | NLinear / DLinear | | KNF | |
|---|---|---|---|---|---|
| | | MAE | MSE | MAE | MSE |
| Traffic | 96 | 0.279 | 0.410 | **0.254** | **0.158** |
| | 192 | 0.284 | 0.423 | **0.276** | **0.176** |
| | 336 | **0.290** | 0.435 | 0.301 | **0.197** |
| | 720 | **0.307** | 0.464 | 0.344 | **0.247** |
| Exchange | 96 | **0.203** | **0.081** | 0.234 | 0.095 |
| | 192 | **0.293** | **0.157** | 0.331 | 0.175 |
| | 336 | **0.414** | **0.305** | 0.456 | 0.327 |
| | 720 | 0.601 | 0.643 | **0.589** | **0.517** |

Table 8: Prediction MAE and MSE on the Traffic and Exchange Benchmark Datasets. KNF achieve MAE scores comparable to the SOTA scores and significantly better MSEs on the traffic dataset.

