# OpenReview forum: "Koopman Neural Operator Forecaster for Time-series with Temporal Distributional Shifts"
_ICLR.cc/2023/Conference — ICLR 2023 poster_

### Official Review · Reviewer_BYPh · 2022-10-21

**Confidence:** 4
**Correctness:** 3
**Technical Novelty And Significance:** 3
**Empirical Novelty And Significance:** 3
**Recommendation:** 6

**Clarity, Quality, Novelty And Reproducibility:**

In general I found this a potentially valuable contribution, depending on whether my understanding of some of the details is correct. I also liked the interpretability aspect of the KNF.

1) The presentation of the model, incl. Fig. 1, I didn’t find very clear, and so I’m not sure I understood everything correctly. For instance, in eq. 1 the problem is stated as one of learning a dynamical system, but right below in eq. 2 it’s then formulated as essentially a time series (auto-) regression problem, which is a quite different objective. My understanding is the authors compile a library of basis functions (similar as in Brunton et al. 2016 pnas), the parameters of which and their weights are learned through the Transformer encoder-decoder? If correct, this should be made more explicit and formal (I didn’t really find this information, incl. the specific type of parameterization and the complete library used, in the text). I also didn’t quite get how (and why?) the time series were transformed into a latent code (just an affine transformation?), and how these parameters were learned. So sect. 3.2 I think needs to be made *much* clearer with all mathematical details on the transformations used and the specific parameterizations of learned functions, what exactly the output of the encoder is etc. etc. Fig. 1 should also be improved.
Much more details about hyper-parameter tuning for all models are also necessary I think. Also, I didn’t find any statement on code availability (did I miss it)?

2) In general I found the anchoring in Koopman operator and dynamical systems theory only very loose at best, and the model is not really geared toward extracting governing equations from the data, unlike much of the recent efforts in ML for dynamical systems (e.g. [1-3]). And I don’t think it needs to be, it’s fine, in my mind, if it just takes inspiration from these fields. So I would suggest to simply tune down and clarify any claims in this direction, i.e. introduce this work as what it is, namely an auto-regressive model for non-stationary time series forecasting which is based on just a library of fixed basis functions. If the authors want to make any stronger (theoretical) claims about their model, e.g. in direction governing equations, there is much more they need to show in my mind (see [1-3]), as well as different models to compare to!

3) This is important: The authors include a local feedback loop, which is similar in spirit to a Kalman filter correction in my understanding. Fig. 4 shows this component is essential for good predictions adapting to the non-stationarity. If I got this right, this component is actually adjusted based on the forecasting time window? But isn’t this then basically violating the basic training-test split scenario, as part of the model gets re-adjusted within the forecasting horizon? At the very least, isn’t this giving the author’s model an unfair edge over all the other competitors which do not use such local information?

4) Baseline models: This paper doesn’t develop novel theory but focuses on engineering a good forecasting model based on, essentially, known components and techniques. For instance, sliding window approaches are very common for non-stationary data, the feedback loop is in my understanding just a variation of the nonlinear Kalman filter theme, and the ‘Koopman operator’ part here basically boils down to the classic technique of providing the model with a set of fixed basis functions (and perhaps all these aspects should be pointed out in a revision!). So I really see the main strength of this contribution in the development of an architecture that outperforms other state-of-the-art models on challenging real-world benchmarks. At the same time, for the same reasons, I would have liked to see a few more comparisons:
- I believe it’s really important to also compare to very simple baselines, like ARIMA or just moving averages or kNN for time series. In my mind there is generally a tendency to develop overly complicated models in DL without properly checking whether very simple and naive methods could already do the job.
- I’m not quite sure why different models have been used as competitors for the two sets of time series in Table 2 and Table 3? Why not apply the same set of models to all?
- LEM [4] in my mind is another strong candidate that should have been tested, and perhaps also Informers. Less important, but just for curiosity I would have been interested how models perform actually designed for extracting governing equations (like those in [1-3]), although not build for non-stationary time series prediction.

5) Presentation of data: Table 2 needs to be in the main text in full (i.e., Table 7). As it’s presented now, this is really cherry-picking of the results where the own model outperforms others. Also I find Tables (like Table 2) which do not include any indication of standard error essentially meaningless from a statistical point of view. It is pretty clear in my mind that there will always be some bias toward the own model, let alone because authors simply know their own model best, and so at the very least an indication of significance needs to be provided. I would furthermore suggest to also indicate the % improvement over the best competing method.

Minor things:
- p.6: In my view forecastability should be based on Lyapunov exponents (not on a linear decomposition like Fourier spectrum), since these are really what determines the forecast horizon in a dynamical system.
- p.9: It’s clear almost any model will perform well on a simple oscillator. But most real world data are likely chaotic, and so these would constitute a more interesting test case (e.g. [5]).

[1] https://arxiv.org/abs/2106.06898

[2] https://arxiv.org/abs/2207.02542

[3] https://arxiv.org/pdf/2201.05136

[4] https://arxiv.org/abs/2110.04744

[5] https://arxiv.org/abs/2110.07238


**Strength And Weaknesses:**

Strengths:
- superior prediction of non-stationary time series
- interpretability of model

Weaknesses:
- only loose anchoring in theory (see below)
- some parts unclear (see below)


**Summary Of The Paper:**

The authors introduce a new architecture for forecasting non-stationary time series, the ‘Koopman Neural Forecaster (KNF)’. KNF combines local and global Koopman matrix operators assembled from a set of basis functions, and also includes a feedback operator based on local prediction errors. Parameters of the Koopman operators are learned through a Transformer encoder-decoder network. The KNO is compared to several SOTA baseline models on challenging, non-stationary real-world time series from the M4 dataset as well as Crypto and basket ball player trajectories.

**Summary Of The Review:**

Potentially nice contribution with superior prediction for non-stationary time series (an important problem), and a model that can in part be interpreted. In my mind somewhat weak anchoring in theory, although not essential here I think. Some more baselines should be considered and presentation of results should be improved.

---

> ### Author Response · Authors · 2022-11-16
> **Response to Reviewer BYPh-Part one**
>
> Thanks so much for your comments.
>
> $\textbf{Q0:}$ This is important:...But isn’t this then basically violating the basic training-test split scenario, as part of the model gets re-adjusted within the forecasting horizon? At the very least, isn’t this giving the author’s model an unfair edge over all the other competitors which do not use such local information?
>
> $\textbf{A0:}$ There might be a misunderstanding of how the feedback loop operates. The feedback module is only based on measurements on the lookback window. We generate the forecasts in an autoregressive way. For instance, to generate 30-ahead predictions, we can use our model to generate 5-step predictions at each call and feed the model’s own predictions back to the input and drop the oldest five input observations. Hence, it does not violate the train-test split scenario. That means the local operator and the adjustment operator depend on time $t$ and vary across different autoregressive calls, adapting to the distribution changes along prediction.
>
> We note that our model has the same information as all other baselines and all comparisons in our paper are fair.
>
> $\newline$
>
> $\textbf{Q1:}$ The presentation of the model, incl. Fig. 1, I didn’t find very clear…If correct, this should be made more explicit and formal…
>
> $\textbf{A1:}$ Thanks for the suggestions - we’ve improved the presentation of the model.
>
> Your understanding is correct. By learning dynamical systems, we do not mean discovering the governing equations but finding a model that can accurately predict the evolution of dynamics. We use the MLP encoder to learn the coefficients of the measurement functions and the Transformer encoder to learn the local operator.
>
> Sec. 4.5 describes the measurement functions that we used: polynomials up to the fourth order, one exponential function, and trigonometric functions with the same number of input steps for each feature.
>
> $\newline$
>
> $\textbf{Q2:}$ …So sect. 3.2 I think needs to be made much clearer with all mathematical details on the transformations used and the specific parameterizations of learned functions…
>
> $\textbf{A2:}$ We’ve rewritten section 3.2.  The details of the hyper-parameter tuning are in Appendix A.3 and Table 6.
>
> $\newline$
>
> $\textbf{Q3:}$ Also, I didn’t find any statement on code availability (did I miss it)?
>
> $\textbf{A3:}$ It is right before the reference section. We will release the code after the review process.
>
> $\newline$
>
> $\textbf{Q4:}$ In general I found the anchoring in Koopman operator and dynamical systems theory only very loose at best…
>
> $\textbf{A4:}$ Thanks for bringing these to our attention. As you mentioned, our method is designed to improve forecasting accuracy for non-stationary time-series, rather than learning governing equations from the data. Note that our modules are based on deep neural networks and KNF does not yield a simple equation to describe the data.
>
> $\newline$
>
> $\textbf{Q5:}$ I believe it’s really important to also compare to very simple baselines, like ARIMA or just moving averages or kNN for time series….
>
> $\textbf{A5:}$ That’s a great suggestion!
> We’ve added the vector autoregressive (VAR) model, which is an extension of ARIMA for multivariate time series, as an additional baseline for the Cryptos and Trajectory datasets. The results are included in the updated version Table 3.
> We trained two types of vector autoregressive models, VAR-local and VAR-global. VAR-global is a single model trained on the entire training set. The VAR-local is to train a VAR model on the lookback window for every test sample. Since VAR-local is significantly worse than VAR-global and other models, we only include the results of VAR-global in the updated version.
>
> $\newline$
>
> $\textbf{Q6:}$ I’m not quite sure why different models have been used as competitors for the two sets of time series in Table 2 and Table 3? Why not apply the same set of models to all?
>
> $\textbf{A6:}$ The M4 dataset is from M4 Competition [1] and is a public benchmark dataset. In this competition, many different methods have been tested, from statistical methods (such as ARIMA and exponential smoothing) to deep learning (such as LSTM and MLP), and SOTA hybrid methods.  More information can be found in [1]. The main paper and the appendix include all the details about the results.
> We have focused on outperforming the best-performing models reported to date on M4, in a fair way.
> The Cryptos and Trajectory datasets are not public benchmarks. So we chose different types of SOTA baselines for time series and ran the baselines by ourselves.
>
> [1] Makridakis et. al; The M4 Competition: 100,000 time series and 61 forecasting methods.

---

> > ### Author Response · Authors · 2022-11-16
> > **Response to Reviewer BYPh - Part Two**
> >
> > $\textbf{Q7:}$ LEM in my mind is another strong candidate that should have been tested…
> >
> > $\textbf{A7:}$ LEM is not designed for time series forecasting and has never been used on forecasting tasks before. It focuses on modeling long-term dependency instead of coping with nonstationarity. It could be possible to generalize the high level ideas from it for time- series forecasting, but it would not be a trivial modification.
> > We’ve included FEDformer as our baseline. It has been shown the best-performing Transformer-based model on time series forecasting [2], so we don’t think it is necessary to include the Informer as well. More importantly, transformer-based models have been shown not effective for time series forecasting [3].
> >
> > [2] Zhou et.al; FEDformer: Frequency Enhanced Decomposed Transformer for Long-term Series Forecasting.
> >
> > [3] Zeng et.al; Are Transformers Effective for Time Series Forecasting?
> >
> > $\newline$
> >
> > $\textbf{Q8:}$ Presentation of data: Table 2 needs to be in the main text in full (i.e., Table 7). As it’s presented now, this is really cherry-picking of the results where the own model outperforms others.
> >
> > $\textbf{A8:}$  Thanks for the suggestion. Our goal is not to cherry-pick, but to focus on the regimes that we design our method for.
> >
> > Note that our focus is on highly non-stationary time series with temporal distributional shifts that are difficult for deep learning to predict.  We chose M4-monthly, M4-weekly, and M4-daily datasets because they are much more difficult to predict than the other relatively stationary M4 datasets, as shown in Table 1.
> >
> > We demonstrated that our model achieved SOTA performance on them in Table 2, and in the meanwhile, we show our model can achieve comparable performance on the remaining M4 datasets.
> >
> > We also note that no model seems to achieve the best performance consistently across all six M4 datasets.
> >
> > $\newline$
> >
> > $\textbf{Q9:}$ Also I find Tables (like Table 2) which do not include any indication of standard error essentially meaningless from a statistical point of view…
> >
> > $\textbf{A9:}$ Sec. 4.5 describes the evaluation setup. We follow the evaluation rules of the M4-competition which only uses ensemble mean.. Ensembling is used by most models in the M4-competition and N-Beats (Oreshkin et al., 2019), so we ensemble five KNF with best hyperparameters but trained with random seeds. Since the evaluation in the M4 competition is only based on a single submission, we also report the sMAPE of a single ensemble prediction without standard deviation.
> >
> > We’ve added the percentage of improvement over the best competing method in Table 2.
> >
> > $\newline$
> >
> > $\textbf{Q10:}$ In my view forecastability should be based on Lyapunov exponents (not on a linear decomposition like Fourier spectrum), since these are really what determines the forecast horizon in a dynamical system.
> >
> > $\textbf{A10:}$ That’s a great point. Using Lyapunov exponents is an good idea but it’s hard to compute  in practice as it often requires  the exact dynamical system/governing equations and initial conditions. To measure the Lyapunov exponents of time series without known governing law, some works, e.g. [4], fit a model first and then compute the Lyapunov exponents based on the fitted model. But this renders the calculation of  Lyapunov exponents dependent on the choice of the model. Instead, our goal was to propose a metric that does not depend on modeling assumptions much.
> >
> > [4] Shintani et al. Is there chaos in the world economy? A nonparametric test using consistent standard errors.

---

### Official Review · Reviewer_ab4B · 2022-10-24

**Confidence:** 4
**Correctness:** 3
**Technical Novelty And Significance:** 3
**Empirical Novelty And Significance:** 3
**Recommendation:** 8

**Clarity, Quality, Novelty And Reproducibility:**

As mentioned above, the method section can be improved. This is also important in terms of reproducibility, which at the current version would be difficult to achieve given that many hyper-parameters are not listed. It would also help if a proper network architecture illustration or table is given, listing all components, network layers, activations, etc. (beyond Fig. 1).

The paper is of high quality in terms of methodology and evaluation setup. To the best of my knowledge the proposed method differs from existing Koopman-based approaches in the measurement functions applied in the latent space (although similar ideas exist in the extended DMD paper and derived work and the decomposition of the Koopman operator to three different objects. In my opinion, these changes are novel and warrant a publication at ICLR. The paper could be stronger if the decomposition could be associated to Koopman theory, however, I do not have a good idea on how to approach this.

**Strength And Weaknesses:**

Probably the strongest point of the paper is the results obtained on several challenging datasets in comparison to several strong baselines. To the best of my knowledge, the particular datasets considered in this work were not previously considered in Koopman-based approaches. Moreover, surpassing strong baselines such as N-BEATS and Smyl on M4 benchmark is particularly encouraging and highlights the capabilities of the proposed approach.

There are also a few weaknesses. The related work section and the overall discussion regarding previous work on Koopman methods can be improved significantly. For instance, the comment in the conclusion (and abstract): ``... this is the first time that Koopman theory is applied to real-world time series without known governing laws.'' is inappropriate and should be re-phrased. Two examples (that even appeared in ML venues) and deal with sequential data for which governing laws are unknown is "DeSKO: Stability-Assured Robust Control with a Deep Stochastic Koopman Operator" by Han et al. that showed multiple examples of real-world control tasks. The second paper "Learning Compositional Koopman Operators for Model-Based Control" by Li et al. also considered a few real-world control problems. Actually, there is another relevant paper that comes to mind in this context which deals with Koopman-based prediction of ECG signals ("12-lead ECG Reconstruction via Koopman Operators" by Golany et al.). Also, even in papers such as Azencot et al. where the authors claim the tasks were "synthetic time series with known governing equations", examples involving forecasting of sea surface temperature were investigated. While these phenomena generally follow the Navier--Stokes equations, real-world data such as sea surface temperature can be modeled with NS only on its dominant factors, whereas fast and slow dynamical processes are not modeled well. Finally, please change the DMD citation of Brunton et al. 2016 to the correct one ("Dynamic mode decomposition of numerical and experimental data" by Schmid). You may also want to include several Koopman-based papers discussing Koopman with control, given that the paper deals with a basic version of control via a feedback loop.

Another weakness is related to the methodology section. Specifically, the formulation around Eq. (4) could be improved significantly. For instance, there is a $v_i \in \mathbb{R}^k$ which seems like a leftover from a previous formulation. Also, it is not clear to me what you mean by $V_t = \Psi(X_t) X_t$ (left of Eq. (4)). Is it matrix multiplication? is it batch multiplication? The dimensions as they appear in the document do not make sense. It would be good if you can add a discussion describing the differences between training and inference with respect to how the global, local and adjustment operators are being computed and updated. Why do you use an adjustment as a multiplication operator and not add the residual in Eq. (9) to the forecast in E. (8)? The equations (11)-(13) seem to not really support the ansatz in Eq. (7). Specifically, while you compute local and adjustment operators per time step, Eq. (12)-(13) use powers of the operators at the *same time*. Shouldn't it be a product of different operators? Please discuss this point. How do you choose the set of measurement functions? Is the method sensitive to the choice of functions?

**Summary Of The Paper:**

This paper proposes a new deep learning model for long-term time series forecasting with distribution shifts based on Koopman operator theory. In comparison to existing work, the authors utilize measurement functions (e.g. sine functions) in latent space, and they formulate forecasting using global and temporally local Koopman operators, as well as an additional adjustment operator based on a feedback loop. The authors evaluate their approach on three challenging datasets in comparison to several baseline methods, showing several promising results.


**Summary Of The Review:**

In summary, this paper suggests a new and interesting Koopman-based approach for long-term of time series data with distribution shifts. The benchmark results position this method in line with state-of-the-art forecasting (statistical and learning) approaches, which will definitely inspire others to build on the proposed architecture to further improve forecasting capabilities. The exposition shortcomings could be handled during the revision period, and thus should not prevent publication.

---

> ### Author Response · Authors · 2022-11-16
> **Response to Reviewer ab4B**
>
> Thanks so much for your reviews
>
> $\textbf{Q1:}$ … The related work section and the overall discussion regarding previous work on Koopman methods can be improved significantly. For instance…is inappropriate and should be re-phrased.
>
> $\textbf{A1:}$  Thanks for pointing that out. We have modified that.
>
> $\newline$
>
> $\textbf{Q2:}$  …the formulation around Eq. (4) could be improved significantly. For instance, there is a  vi∈Rk, which seems like a leftover from a previous formulation. Also, it is not clear to me what you mean by  Vt=Ψ(Xt)Xt…
>
> $\textbf{A2:}$  The operation between $\Psi (X_t)$ and $X_t$ is a composition of element-wise product and summation over the last axis, as shown below.
>
> $$ V_t [i,j] = \sum_l \Psi(X_t) [i,j,l]X_t [j, l]; 1\leq i \leq n, \; 1\leq j\leq d, \; 1\leq l\leq k$$
>
> Where $i,j,l$ denote the indices of three axes. It’s equivalent to applying $n$ different linear transformations to the $k$ observations of each feature.  We’ve updated the equations in our revised version.
>
> $\newline$
>
> $\textbf{Q3:}$  …describing the differences between training and inference with respect to how the global, local and adjustment operators are being computed and updated.
>
> $\textbf{A3:}$  There is no difference in how these operators are computed and used between training and inference. The global operator is shared across all samples and it is fixed after training. Both local and adjustment operators are computed by neural nets only based on the measurements on the lookback window.
>
> $\newline$
>
> $\textbf{Q4:}$  Why do you use an adjustment as a multiplication operator and not add the residual in Eq. (9) to the forecast in E. (8)? The equations (11)-(13) seem to not really support the ansatz in Eq. (7).
>
> $\textbf{A4:}$  The global+local operator is already trained to be the best fit matrix for the time series in the look-back window. Therefore, there is no residual in the look-back window. But if there are distributional shifts that happen in the look-back window, a single propagation matrix/rule is insufficient to model multiple distributions. The global+local operator added by the adjustment matrix is still a single matrix.
>
> Thus, we want the feedback module to detect these distributional shifts in the lookback window and use the adjustment matrix to make the global+local operator adapt to the latest distribution in the lookback window. We assume the first few steps in the forecasting window follow the same distribution as the last distribution in the lookback window. That’s why we use the adjustment matrix only during the forecasting window.
>
> This design also helps to deal with distributional shifts in the forecasting window because we generate the forecasts in an autoregressive way. For instance, to generate 30-ahead predictions, we can use our model to generate 5-step predictions at each autoregressive prediction step and feed the predictions back to the input and drop the oldest 5 input observations. That means the local operator and the adjustment operator depend on time $t$ and vary across different autoregressive calls, adapting to the distribution changes along predictions.
>
> $\newline$
>
> $\textbf{Q5:}$  Specifically, while you compute local and adjustment operators per time step, Eq. (12)-(13) use powers of the operators at the same time. Shouldn't it be a product of different operators? Please discuss this point.
>
> $\textbf{A5:}$  The $t$ denotes the time of the last input step. Since we generate forecasts in an autoregressive manner, the $t$ as well as the local and adjustment operators change across different autoregressive steps.
>
> $\newline$
>
> $\textbf{Q6:}$  How do you choose the set of measurement functions? Is the method sensitive to the choice of functions?
>
> $\textbf{A6:}$  The measurement functions are common basis functions in [1,2,3]. The low complexity functions, such as polynomials and sinusoidals,  are commonly reasonable choices across tasks unless the dynamics are very specific. In practice, we have an ablation study of the measurement functions on an M4 dataset in the appendix.  All functions we chose have shown benefits for superior accuracy.
>
> [1] Takeishi et al. Learning Koopman invariant subspaces for dynamic mode decomposition.
>
> [2] Arbabi et al. Ergodic theory, dynamic mode decomposition, and computation of spectral properties of the Koopman operator.
>
> [3] Kutz et al. Koopman theory for partial differential equations.

---

> > ### Comment · Reviewer_ab4B · 2022-11-23
> > **Thank you!**
> >
> > Thank you for addressing my comments.

---

### Official Review · Reviewer_63Af · 2022-10-25

**Confidence:** 4
**Correctness:** 3
**Technical Novelty And Significance:** 2
**Empirical Novelty And Significance:** 3
**Recommendation:** 6

**Clarity, Quality, Novelty And Reproducibility:**

The presentation of this paper is clear, leading to an easy understanding. There are only two doubts mentioned above.

**Strength And Weaknesses:**

mentioned above.

**Summary Of The Paper:**

This paper proposed a deep sequence model, KNF, via the Koopman theory for time series forecasting. KNF employs a global operator to learn shared characteristics, and a local operator to capture changing dynamics, which is claimed to impose appropriate inductive biases for improved robustness against distributional shifts. The experiments conducted on several datasets confirm the effectiveness of the proposed model.

The presentation of this paper is clear, leading to an easy understanding. I have two doubts about the claimed contributions
1. The KNF is proposed for handling time series with temporal distribution shifts. Hence, it is natural to verify the performance of KNF using real-world datasets with distribution shifts. As introduced in Subsection 4.2, the conducted Electricity dataset seems to exhibit only seasonality. Does distribution shift refer to periodicity? Thus, I have a doubt that how to identify the time series with temporal distribution shifts, or the killer applications of KNF in practice. Otherwise, it is better to provide certain theory to support this claim.

2. It is also a need to point out the key component of the proposed KNF model that enables translation invariance. Besides, it is better to provide a comparison or discussion with previous "global-local" models, such as [Sen, R., Yu, H. F., & Dhillon, I. S. (2019). Think globally, act locally: A deep neural network approach to high-dimensional time series forecasting. Advances in neural information processing systems, 32.]

Minor issues:
1. What do the bold notes in Table 1 mean?
2. When using KNF to forecast time series, does the user have to guarantee that the size of the forecast window is the same as the size of the lookback window? Is there any trick to setting this window size? What is the impact if the window size is much smaller than the period?

**Summary Of The Review:**

Overall, I tend to accept this paper if the authors fixed my doubts in the next phase.

---- after rebuttal ----

I have read the authors' rebuttal, and I believe the responses have fixed my doubts. Therefore, I tend to accept this paper.

---

> ### Author Response · Authors · 2022-11-16
> **Response to Reviewer 63Af**
>
> Thanks so much for your reviews.
>
> $\textbf{Q1:}$ …As introduced in Subsection 4.2, the conducted Electricity dataset seems to exhibit only seasonality. Does distribution shift refer to periodicity? …
> $\textbf{Q2:}$  What do the bold notes in Table 1 mean?
>
> $\textbf{A1/ A2:}$  There was a misunderstanding. We did not use the Electricity data for our experiment because it is relatively stationary and does not have temporal distributional shifts, which is not our focus. Distribution shifts in our setting refer to the concept shift, which means $P(Y|X)$ (or $P(X_{t+1}|X_t)$ in our setting) changes over time. For instance, $\dot x = ax+b$ with $a,b$ changing over time.
>
> The clear seasonality in the electricity data indicates it does not have many temporal distributional shifts - the time-series datasets with seasonality have predictable repeating patterns while the focus of our paper is on the very challenging task of time-series with evolving distributions.
>
> The bold datasets are the datasets we use in our paper. As shown in Table 1, we use three different metrics, including forecastability, trend, and seasonality, to measure the difficulty to forecast.  The datasets we use, including Crypto, Player Trajectories, M4-monthly, M4-weekly, and M4-daily, have low forecastability, no clear trends, and low seasonality. It indicates these datasets have high nonstationarity and are difficult to predict.
>
> We hope this resolves the misunderstanding about the contributions and the time-series tasks for which our paper shows clear value.
>
> $\newline$
>
> $\textbf{Q3:}$  It is also a need to point out the key component of the proposed KNF model that enables translation invariance.
>
> $\textbf{A3:}$  All the models we included in our paper are translational equivariant since they are applied in an autoregressive way.
>
> $\newline$
>
> $\textbf{Q4:}$  Besides, it is better to provide a comparison or discussion with previous "global-local" models, such as [Sen, R., Yu, H. F., & Dhillon, I. S. (2019). Think globally, act locally…
>
> $\textbf{A4:}$  Sen et al focuses mainly on better modeling the relationships between time series rather than advances in modeling over time. Their global-local refers to the relationships across different covariants, whereas our global-local is on the time dimension.They use global TCN to avoid normalization before training when there are wide variations in scale.  In addition, the experiments only focus on relatively stationary datasets, such as electricity and traffic, which is different from ours.  We’ve added the discussion of this paper in our revised version.
>
> $\newline$
>
> $\textbf{Q5:}$  When using KNF to forecast time series, does the user have to guarantee that the size of the forecast window is the same as the size of the lookback window? Is there any trick to setting this window size? What is the impact if the window size is much smaller than the period?
>
> $\textbf{A5:}$  No - the lookback window size and forecasting window size do not need to be the same. The lookback and forecasting window are hyperparameters that can be tuned based on the validation performance. During inference, we can simply apply the learned operators recursively to generate predictions with any desired length.
>
> The optimal input window size varies across different datasets. For instance, M4-daily only needs an input length of 21 but the optimal input window size on the Cryptos dataset is 63. Since the datasets we use do not have seasonality, it is difficult to determine the length of a period. But it is natural that the prediction performance would deteriorate if the input length is too small due to the lack of sufficient information for prediction.

---

> > ### Author Response · Authors · 2022-12-02
> > **Message to Reviewer 63Af**
> >
> > Dear Reviewer 63Af,
> >
> > We have worked to address each of the points you mentioned. We would very much appreciate it if you could take some time to review our responses and updated paper.
> >
> > Thank you!

---

> > > ### Comment · Reviewer_63Af · 2022-12-06
> > > **Reponse**
> > >
> > > I have read the authors' rebuttal, and I believe the responses have fixed my doubts. Therefore, I tend to accept this paper.

---

### Official Review · Reviewer_CZDy · 2022-10-28

**Confidence:** 3
**Correctness:** 3
**Technical Novelty And Significance:** 3
**Empirical Novelty And Significance:** 3
**Recommendation:** 8

**Clarity, Quality, Novelty And Reproducibility:**

Clarity:

I'm confused about the encoder. Is the output really supposed to be 3D? Is v_i really k-dimensional?

"But computing spectral decomposition for every sample is computationally expensive, so we utilize DNNs to learn the Koopman operator... While the Koopman matrix should vary across samples and time in our case..." I found these uses of "sample" confusing, but eventually figured out that you're using "sample" to mean one trajectory. Maybe that could be stated explicitly. (You could apply DMD to multiple trajectories by stacking them up if you expect one matrix A to be valid across all of the trajectories. However, this is certainly an approximation and, especially in the context of studying temporal distribution shift, it makes sense that just applying DMD once wouldn't work.)


Reproducibility:

The networks seem vaguely described. For example, it's mentioned in the appendix that the hyperparameter tuning range for number of layers is 3-7. Do the encoder, decoder, transformer encoder, and the feedback module lamda all have the same number of layers? We also don't know what the nonlinearity is, etc. However, the authors say they will release the code, so that would help.


Novelty:

The abstract says, "To the best of our knowledge, this is the first time that Koopman theory is applied to real-world chaotic time series without known governing laws," and the conclusion contains a similar sentence. However, this is not the first. For example, see [A].  They use three real-world datasets, including measles outbreak data, and specifically mention that it's been shown that measles outbreaks are chaotic. On the other hand, it is not shown that the datasets in this paper are chaotic.

The main distinction given from previous Koopman papers is "Most of these works use rather simple DNN architectures, and are applied to synthetic time series with known governing equations. Different from these, we focus on the real-world time series with no governing laws, such from finance." Since none of these papers are used as baselines, and previous Koopman papers have certainly been applied to real datasets, I would suggest emphasizing the approach of letting the Koopman matrix evolve over time in a three-piece way (global, local, and feedback loop). As far as I know, this is novel. As a **bonus**, it would be interesting to see to what extent the Koopman matrix evolves. For example, how much influence does the global piece have, or does it get dominated by the others? If there is a clear distribution shift halfway through the data, is that easy to see in the pieces of the Koopman operator?


Quality/Correctness:

"Fig. 6 shows that our model can always make perfect predictions on the test set..." It cannot be true that a deep learning model can "always make perfect predictions" of this system. There has to at least be rounding error. :) What is the error?

"We define a set of measurement functions G := [g1, · · · , gn] that spans the Koopman space..." The Koopman matrix is nd x nd. Is there a way to know that the few measurement functions you picked will span the space?

In A.1.1:
- "It can also be spanned by three eigenfunctions..." There are two listed. Is there a comma missing?
- "our model can always learn the correct eigenfunctions." What do you mean by "correct"? Did you check that the eigenfunctions are the expected ones, or just that the prediction is accurate? I could imagine that the encoder transforms the data and finds some other valid combination that results in low error.

"Table 2 reports the prediction sMAPE... on six datasets with different frequency in M4." There are only 3 datasets in this table. I see from later in the paragraph that the rest were moved to the appendix.

"We run experiments with our model on a simple oscillator system..." This isn't an oscillator system. The eigenvalues are real and the system decays down to an attractor.


[A]  Brunton, S.L., Brunton, B.W., Proctor, J.L. et al. Chaos as an intermittently forced linear system. Nat Commun 8, 19 (2017). https://doi.org/10.1038/s41467-017-00030-8.


**Strength And Weaknesses:**

To my knowledge, this a novel way to use Koopman theory for time series predictions. I think that the comparisons against baselines are impressive, especially when comparing to the recent M4 competition. Table 1, demonstrating that the datasets are difficult to predict, is helpful, and I thought the ablation studies were well-done.

I have some comments in the next box, but none of them are major or difficult to fix. For sure, the claims along the lines of  "To the best of our knowledge, this is the first time that Koopman theory is applied to real-world chaotic time series without known governing laws" need to be fixed.


**Summary Of The Paper:**

This paper is about a new method called Koopman Neural Forecaster (KNF) for forecasting time series that have distributional shifts. It combines three operators: a global one, a local one, and one that can update over time via feedback.

**Summary Of The Review:**

I think that this paper is working on an important problem with broad applications. The approach seems interesting, and the results are impressive. I'll be interested in trying this method out once the paper is published. I have some comments and questions, but they are not major.

---

> ### Author Response · Authors · 2022-11-16
> **Response to Reviewer CZDy-Part one**
>
> Thanks for your insightful comments.
>
> $\textbf{Q1:}$ the claims along the lines of "To the best of our knowledge…" need to be fixed.
>
> $\textbf{A1:}$ We have removed that sentence.
>
> $\newline$
>
> $\textbf{Q2:}$ I'm confused about the encoder. Is the output really supposed to be 3D? Is v_i really k-dimensional?
>
> $\textbf{A2:}$ The encoder output is a one-dimensional vector of length $ndk$ and then reshaped into a 3D tensor for the following operation with $X_t$.  $v_i$ is actually $d$-dimensional and we’ve fixed it. Thanks for catching that.
> We have made this part clearer in the revised version.
>
> $\newline$
>
> $\textbf{Q3:}$ I found these uses of "sample" confusing,... (You could apply DMD to multiple trajectories by stacking them up if you expect one matrix A to be valid… just applying DMD once wouldn't work.
>
> $\textbf{A3:}$ Thank you for your suggestion. A sample means a slice (a continuous subsequence) of a time series. We have clarified this in the revised version.
>
> $\newline$
>
> $\textbf{Q4:}$ The networks seem vaguely described… all have the same number of layers?...
>
> $\textbf{A4:}$ No, they do not have the same number of layers. We tune the number of layers/hidden dimensions of the encoder, decoder, transformer encoder, and the feedback MLP separately with grid search. So they may have different numbers of layers. For example, on the M4 dataset, the encoder and decoder have five layers and the feedback module and the transformer encoder have three layers. As the nonlinearity, we simply use ReLU for all models. We’ve added more details to Appendix A.2.
>
> $\newline$
>
> $\textbf{Q5:}$ I would suggest emphasizing the approach of letting the Koopman matrix evolve over time in a three-piece way … how much influence does the global piece have, or does it get dominated by the others? If there is a clear distribution shift halfway through the data, is that easy to see in the pieces of the Koopman operator?
>
> $\textbf{A5:}$
> Thanks for your suggestion. We’ve rephrased that and emphasized our novelty in the design of the Koopman operator in our introduction.  Actually, in the ablation study, the KNF-base-G can be considered as a classic Koopman autoencoder used in previous studies [1,2].
> [1] Yeung et al. Learning deep neural network representations for Koopman operators of nonlinear dynamical systems.
> [2] Lusch et al. Deep learning for universal linear embeddings of nonlinear dynamics.
>
> Adding the global operator improves the prediction performance (8.435->7.500) based on the ablation study. The intuition behind this is that the global operator is shared across all samples and thus has low variance. On the contrary, though the local matrix could evolve over time, it may have high variance since it is only computed based on the measurements in the lookback window of a sample. Especially when the number of measurements on the lookback is smaller than the Koopman space dimension, there is no unique solution for the Koopman matrix. In other words, both operators are essential since the local operator enables the Koopman matrix to be time-varying and the global operator improves robustness.
>
> The adjustment operator's norm can be used to measure the effects of the distributional shifts.
> We leave this exploration to future work.
>
> $\newline$
>
> $\textbf{Q6:}$ "Fig. 6 shows that our model can always make perfect predictions on the test set..." It cannot be true that a deep learning model can "always make perfect predictions" of this system. There has to at least be rounding error. :) What is the error?
>
> $\textbf{A6:}$ We have modified that sentence. The prediction RMSEs corresponding to the four different dictionaries (D1 ~ D4) are 1.915e-05, 1.328e-05, 5.274e-05, and 6.626e-05.
>
> $\newline$
>
> $\textbf{Q7:}$ The Koopman matrix is nd x nd. Is there a way to know that the few measurement functions you picked will span the space?
>
> $\textbf{A7:}$ This is an important point and discussed in Sec. 4.5. We use polynomials up to the fourth order, one exponential function, and trigonometric functions with the same number of input steps for each feature, and pairwise product interaction functions between features if the time series data is multivariate. Unfortunately, there is no guarantee that the measurement functions will span the space.  However, in practice, it is a good idea to select multiple measurement functions that can cover a wide range of dynamics.

---

> > ### Author Response · Authors · 2022-11-16
> > **Response to Reviewer CZDy-Part two**
> >
> > $\textbf{Q8:}$ "It can also be spanned by three eigenfunctions..." There are two listed. Is there a comma missing?
> >
> > $\textbf{A8:}$ Thanks for catching that. We’ve fixed it.
> >
> > $\newline$
> >
> > $\textbf{Q9:}$ "our model can always learn the correct eigenfunctions." What do you mean by "correct"? … the encoder transforms the data and finds some other valid combination...
> >
> > $\textbf{A9:}$ We’ve clarified this and added a plot of eigenfunctions (Fig. 6 left) in the Appendix in the revised version. Since the learned Koopman matrix may not be unique given the flexibility in the coefficients of measurement functions, and the compositions of eigenfunctions are also eigenfunctions, there are many possible eigenvalues. In Fig. 6, we plot some possible true eigenvalues and we can see that most of the learned eigenvalues still match the true eigenvalues.

---

### Author Response · Authors · 2022-11-16
**General Response**

We thank the reviewers for their insightful comments and thoughtful feedback. In particular, both reviewer CZDy and ab4B agree that our proposed method is novel and the experimental results are strong and promising; Reviewer 63Af pointed out that the presentation of our paper is clear, and Reviewer BYPh appreciates the interpretability of our model. We address the individual questions of each reviewer below.

* We have revised the paper (text in blue). Below are some of the highlights of our revision:
* We have rewritten section 3.2 and clarified the operation in Eqn.4.
* We have tuned down some of our claims as requested.
* We have added an additional baseline, the vector autoregressive model, for Cryptos and Trajectory datasets.
* We have added a visualization of learned eigenvalues for the synthetic experiment in Appendix A.1.

---

### Decision · Program_Chairs · 2023-01-20

**Decision:**

Accept: poster

**Justification For Why Not Higher Score:**

This is a solid paper that cleverly combines existing ideas but does really not introduce new concepts. The stronger point is the experimental results obtained on several challenging datasets. This does not deserve in my opinion a spotlight, but I have nothing against.

**Justification For Why Not Lower Score:**

original contribution and strong results on challenging datasets

**Metareview: Summary, Strengths And Weaknesses:**

The paper proposes a new model, based on Koopman operator theory for forecasting time series with distributional shifts. The method combines global operators shared by time series, local ones that adapt in time for each series, together with a feedback loop for abrupt changes. The evaluation is performed on challenging datasets and shows promising results.

The reviewer agree that the paper is solid and presents new ideas to use Koopman theory for time series prediction. In particular, the authors introduce a novel way to model measurement functions and apply them in a latent space. The reviewers highlight the quality of the results obtained on challenging benchmarks which allows the method to surpass well established SOTA methods. Several reviewers qualify these experimental results as impressive. The responses and modifications during the rebuttal period were acknowledged as satisfying by the reviewers.

**Note From Pc:**

if the above contains the word "oral" or "spotlight" please see: "oral" presentation means -> notable-top-5% and "spotlight" means -> notable-top-25%. As stated in our emails, we are disassociating presentation type from AC recommendations